# Motivational factors were more important than perceived risk or optimism for compliance to infection control measures in the early stage of the COVID-19 pandemic

**Bjørn Sætrevik**[ORCID]\*, **Sebastian B. Bjørkheim**

Operational Psychology Research Group, Department for Psychosocial Science, Faculty of Psychology, University of Bergen, Bergen, Norway

\* bjorn.satrevik@uib.no

## Abstract

Compliance to infection control measures may be influenced both by the fear of negative consequences of a pandemic, but also by the expectation to be able to handle the pandemic's challenges. We performed a survey on a representative sample for Norway (N = 4,083) in the first weeks of the COVID-19 lock-down in March 2020. We had preregistered hypotheses to test the effect of optimism and perceived risk on compliance. Perceived risk had small effects on increasing compliance and on leading to more careful information gathering. The expected negative association between optimism and compliance was not supported, and there was instead a small positive association. We found a small effect that optimism was associated with seeing less risk from the pandemic and with a larger optimistic bias. Finally, an exploratory analysis showed that seeing the infection control measures as being effective in protecting others explained a substantial proportion of the variation in compliance. The study indicates that how we think about pandemic risk has complex and non-intuitive relationships with compliance. Our beliefs and motivations toward infection control measures appears to be important for compliance.

## Introduction

### Background

For a society to successfully handle a pandemic event, most of the population must comply with infection control measures, which may involve changing their daily habits for work, commute and commerce, keeping physical distance and avoiding social events, following hygiene advice and using facemasks. Previous research on pandemic behaviour [1, 2] has often argued that seeing the pandemic as a risk may be necessary to motivate compliance with such measures. Along this line of thinking, estimating the risk to be low may lead to a sense of complacency and insufficient motivation to comply. On the other hand, one may consider a sense of optimism to be necessary to instil a drive to persevere with complying with the measures in

**Funding:** The current manuscript has been prepared as part of the PANDRISK research project, which is funded by a grant awarded to author BS from the Trond Mohn Foundation, project number TMS2020TMT08. More information about the funder can be found here: https://mohnfoundation.no/?lang=en The founder has not played any part in study design, data collection and analysis, decision to publish, or preparation of the manuscript.

**Competing interests:** The authors have declared that no competing interests exist.

times of adversity. There is thus potential for a complex relationship between perceived risk, optimism and compliance.

**Perceived pandemic risk.** Objectively speaking, "risk" is the product of the probability of an event to occur on the value of the event's outcome. The concept of "perceived risk" takes into account the ways human evaluation of both probability and value can deviate from an objective assessment of risk [3, 4]. Perceived risk has been examined from cognitive [5], social [6] and cultural perspectives [7]. In particular, a number of heuristics and biases have been shown to influence how humans evaluate risk [8–10].

In the context of a pandemic [1, 11], perceived risk may express the public's impression of the probability that they will be infected, and their impression of the consequences it would have for them if they were infected (i.e., whether an infection would lead them to develop the disease, and how serious it would be for them to have the disease). Perceived risk could also express the public's impression of the probability for, and consequences of people close to them being infected, or various societal outcomes such as lockdown and economic hardships. A crucial assumption in research on complying with infection control measures [12, 13], has been that individuals are more likely to comply if they feel threatened by the disease. It has been assumed that the threat of being personally affected by the pandemic motivates individuals to take action to prevent being infected. If they do not see the overall threat from the pandemic as being particularly high, or that they do not see it as a relevant threat for themselves, they would have less reason to comply. The core assumption is thus that self-interested concern for your own well-being is a central motivation for choosing to comply. If a sense of threat is crucial for compliance, this would justify a public health approach that emphasizes the personal risk from the pandemic. However, variation in perceived risk may not be a crucial factor to explain variation in compliance in all pandemic settings, and other factors may be of equal or larger importance.

**Optimism in managing the pandemic measure.** The trait of dispositional "optimism" expresses a tendency to expect favourable outcomes from most situations one encounters in life [14]. Optimism is typically seen as a beneficial trait, and has been found to be associated with physical and mental health, and has effects on exerting effort to reaching your goal and persevere in motivation [15–17]. A recent study [18] showed that optimism was positively associated with well-being during the COVID-19 lockdown in Israel. It could be that optimism contributes the drive and motivation to comply with infection control measures over time. If we see little hope of surviving the pandemic, or we do not believe that our intentions or actions will have any impact on the level of risk, we may be less motivated to comply. Such use of the term optimism relates it to factors such as self-efficacy [19] or "coping expectation" [20]. This also relates to the theory of planned behaviour [21], where it is assumed that the attitude of whether compliance will be effective, and the perceived behavioural control in being able to comply will determine the intention to comply.

Specific to the context of a pandemic, optimism may refer to an expectation of being able to handle the various challenges the pandemic may bring. Optimism is likely to be related to perceiving the risk to be lower, but optimism may also have orthogonal effects in how we expect to handle a given risk. Thus, two people with relatively similar assessment of the pandemic risk, may differ in how optimistic they are about being able to handle the challenge (i.e., both may see the probability of infection to be fairly low, and the consequences to their health to be serious, but a more optimistic person may imagine themselves better able to live with the risk or to manage living with the adverse outcome). Optimism may apply to expectations of handling events such as becoming infected, experiencing those close to us being sick or at risk, having to adapt to various infection control measures, having economic hardships, or the psychosocial consequences of living in a dramatically changed world.

While optimism may have positive effects for health and well-being in general, it may have disadvantages for personal safety when a real threat is present. Being overly optimistic may lead to underestimating the danger and not taking sufficient precautions. Optimism could thus lead to lower compliance to infection control measures, which may increase the risk for infection for both self and others.

Further, general optimism may also be related to a type of fallacy in risk perception has been called an optimism bias [22], leading to biased assessment of available information. Van der Pligt [13] highlighted a pattern of results where people see a risk to apply more to others than to oneself. Having an optimistic bias when thinking about the pandemic could reduce compliance with infection control measures, as we think that we are relatively safe from infection, while this risk mainly threatens others. Thus, while dispositional optimism may have overall benefits in everyday life, it may be detrimental during a real threat such as a pandemic, as it lowers compliance [as shown for other health related advice, 23].

As suggested above, optimism can have both positive and negative effects for health and well-being, and it may have particular effects for risk assessment and motivation factors during a specific threat such as a pandemic. The literature is not yet clear on this, and a recent study on a convenience sample in Switzerland [24] did not find an association between optimism and compliance. In the current manuscript we will explore these potentially complex relationships between optimism and compliance during the initial stage of a pandemic.

**Compliance to infection control measures.** Compliance implies responding favourably (i.e., submitting) to an explicit or implicit request made by others. Factors such as social or economic power, authority, trust in expertise or social influence may increase compliance. Here we are interested in compliance in terms of being willing to follow the infection control measures prescribed by official health authorities. This may be actions such as avoiding social gatherings and keeping distance to others, washing hands frequently and thoroughly, limit traveling and wearing facemasks in public. To comply with these measures may constitute relatively large changes to one's way of life, may cause inconvenience or discomfort, and may preclude other actions one would otherwise take (e.g., being socially active or travelling). As such, choosing whether to comply, or the extent of compliance will be weighed against one's other priorities and values. As mentioned above, previous research on behavioural intentions to comply with infection control measures have tended to emphasize the role of perceived risk [12, 13]. However, from a perspective of making choices of whether or not to comply, it may also be relevant to examine the motivation people have to comply with the measures.

According to the theory of planned behaviour [21] the motivation for and likelihood to actually comply with infection control measures will be influenced by behavioural intentions to do so. Intentions are in turn caused by positive attitudes to the measures, believing that significant others expect one to comply with the measures, and feeling that one will be able to comply with the measures. Breckler [25] considered attitudes to consist of affect, behaviour and cognitions. In this sense, the behavioural intentions to comply with the measures may constitute one aspect of the attitude towards the infection measure, while the other components are the affective evaluation of the measures, and the beliefs, thoughts and attributes one holds about the informational value of the recommendation to follow the measures. A crucial belief about the infection control measures may be whether one considers them to be effective in limiting the pandemic. Such beliefs may differentiate between whether one thinks the measures are effective in protecting oneself, or effective in protecting the public in general.

It has been suggested that empathy and prosocial motivation (as opposed to self-interested motivation) may play a larger role in compliance to infection control measures than previously assumed [26, 27]. Thus, one may be motivated to comply not only by wanting to avoid becoming sick oneself, but also by trying to prevent causing others to become sick.

**Information gathering about the pandemic.** During a pandemic that has a profound impact on most aspects of their lives, people tend to be invested in learning as much as possible about the constantly changing aspects of the pandemic. This may include learning about biomedical conditions of the virus and the disease, about local and global current rate of infections, hospitalizations and fatalities, about current and planned infection control measures and vaccination schemes, and about the political and economic consequences of the pandemic.

Public health management of a pandemic relies on the population being informed about the infection mechanisms and the current infection control measures. A study on the H1N1 pandemic [28] found that attention to pandemic news predicted compliance with public health measures. Another study from that pandemic [29] found that feeling sufficiently informed about influenza vaccination was associated with taking the vaccine. A study of convenience samples in Serbia and Latin-America from April 2020 [30] showed that optimism was associated with lower perceived risk and higher compliance.

The public can be assumed to be motivated to attend to and to seek out information about the pandemic to keep healthy and to preserve the health of others. However, the public are confronted with a complex and dynamic situation, where information about the pandemic may come from different sources of variable reliability, the information's validity may change over time, and the public may distrust the information or interpret it differently than intended. The concept of "information gathering" has been used to describe strategies that the general public may use to learn about the risks and orient their behaviour in the face of health threats. How information is gathered and used is influenced by factors such as credibility, trust, social norms, moral messages, message format, heuristics, mental models, and risk comparisons [31–33]. For the current setting, we may expect those who see the pandemic as a threat will be more deliberate in their information gathering about the disease and the infection control measures. Such deliberation may take the form of being careful about which sources of information about the pandemic one considers to be credible, and to trust the information coming from those sources.

## Research needs

As mentioned above, previous studies [28, 34] have indicated that perceived risk may be crucial for compliance with infection control measures. However, these studies tend to be made in hypothetical cases, or for infections that the public may not see as very severe (seasonal influenza or the N1H1 pandemic), and the infection control measures are not considered to be very demanding (e.g., to take a vaccine when it is offered). In contrast, at the time of the current data collection (March 2020), the COVID-19 pandemic was frequently reported as being a severe threat. The Norwegian infection control measures at that time were considered to be extensive and invasive, and were referred to as "the strongest-ever peacetime measures" [35]. It could thus be of interest to investigate the relationship between perceived risk and compliance in a setting where it is likely that most of the population perceives the risk to be quite high and compliance to be costly.

Much of the previous research on compliance to infection control measures has been done on convenience samples, on self-recruited samples, or samples invited with the explicit purpose of studying pandemic behaviour [i.e. 11 in Germany, 36 in the US, 37, in Norway]. Associations identified in such non-representative samples may deviate from the general population in crucial ways. Further, variation in risk evaluation or compliance may have systematic relationships with the willingness to answer surveys on the topic. For example, those who view the risk as high may be motivated to participate in a study in order to convince

others of the threat. Another example is that those with low compliance may not want to participate in a study where they may be confronted with their lack of compliance. Representative samples may be necessary to get a true measure of the public's compliance with infection control measures, and how it is associated with factors such as perceived risk or optimism.

It may also be of value to examine measures taken relatively early in what turned out to be an extensive pandemic, in order to identify the initial conditions for how the public opinion and behavior later developed. At the time of writing the pandemic has lasted for over two years, and has gone through a number of phases and changes in character. An increasing number of factors is likely to contribute to perceived risk and compliance as a pandemic endures, and it could be that the relationships are easier to disentangle in the first stages of a pandemic. There are also other efforts in studying the early stages of the COVID-19 pandemic [38 in Italy, 39 in the US].

Finally, the Scandinavian countries are sometimes described as societies that are high in trust and social cohesion [40]. This may present a special case for compliance to infection control measures, and should be compared to similar studies in other contexts. It should be examined to which extent the previously identified relationships hold also in this context.

## Current study

The current study is part of the overall PANDRISK project (https://www.uib.no/en/pandrisk), which aims to examine the causes and effects of individual variation in perceived risk in the Norwegian population during the COVID-19 pandemic. The current paper will test confirmatory and exploratory hypotheses about associations between optimism and perceived risk, and between perceived risk and compliance with infection control measures. The data are from a nationally representative survey conducted in Norway in March 2020. The preregistration (available at https:/osf.io/umgnr) was made public on March 27 (while the data collection was ongoing, but before any of the data had been made available to the researchers). Note that a previous publication presented more detailed item description, response distributions and visualization [41] from the same dataset. For brevity we will limit the descriptive reporting in the current paper, and instead encourage any reader interested in more detailed description of results to examine our previous paper.

## Hypotheses

The current study measured responses that may be interpreted to indicate optimism, perceived risk, compliance and information gathering. Previous research provided reasons to expect specific relationships between these variables. There is nevertheless freedom to interpret the measurements and relationships in different ways. We therefore preregistered the following hypotheses:

**Effects of optimism.** We assume that people that have an optimistic outlook (in general and on the pandemic in particular) will also tend to see the pandemic risk as lower. We thus expect (H1a) "Optimism" to be negatively associated with "Perceived risk".

Individuals with an optimistic outlook may see themselves to be at lower risk, even if they are exposed to situations with potential contagion. We thus expect (H1b) an interaction of low "Optimism" and high "Exposure" on "Perceived risk".

Further, we expect to see indications of an optimism bias, in terms of seeing oneself to be less at risk for infection than the average Norwegian citizen is. This pattern may be particularly pronounced for individuals that have optimistic outcome expectations of the pandemic. We thus expect (H1c) "Optimism" to be associated with "risk for others" being seen as larger than "risk for self".

It can be argued that optimism may lead to believing that one is relatively safe from the pandemic, regardless of one's actions. This sense of complacency may counteract the motivation to comply with infection control measures. We thus expect (H1d) "Optimism" to be negatively associated with "Compliance".

**Effects of perceived risk.**   As reviewed above, individuals who consider there to be a severe threat of being infected by the pandemic may take more precautions to prevent infections for self and others. We thus expect (H2) "Perceived risk" to be positively associated with "Compliance". The preregistration also suggested a hypothesis H2b for testing whether the level of trust in official health advice had increased compared to before the onset of the pandemic. However, this item was not measured in the same sample as the other items described in this article. Hypothesis H2b will thus not be further discussed in the current paper.

**Effects on information gathering.**   Individuals that see the pandemic as threatening may also be more careful about how they gather information about the pandemic. This may have different effects on the information gathering. More careful information gathering may be expressed as trusting the official information about the pandemic. More careful information gathering may also lead to evaluating information sources in terms of their trustworthiness. We therefore expect (H3) "Perceived risk" to be positively associated with "Information gathering".

**Effects on physical health.**   In addition to the items and hypotheses discussed above and in Table 1, our survey and preregistration also included items about the extent to which participants had been infected, sick or quarantined due to the COVID-19 pandemic. Hypotheses H4 and H5 were planned in order to test associations between "Perceived risk", "Exposure" and "Compliance". However, when the infection rates in Norway for the week of data collection showed that the pandemic at that time was less severe than what was feared when designing our study and hypotheses. This resulted in quite few participants indicating positive answers on these items, and those that reported to have been infected should be interpreted with some caution. Ten people in our sample (.08%) stated to have been confirmed to be infected by the coronavirus by a medical test, and an additional 307 people (2.55%) stated that they assumed they had been infected but had not had this confirmed by testing. The majority (65%) of those who assumed they had been infected stated that they had come down with a stuffy nose and sore throat, but not a high fever.

Given that mild symptoms were difficult to discern from other respiratory infections at the time of data-collection, it is difficult to say whether the 122 participants that assumed they had been infected by the coronavirus, had in fact been so. Only four participants stated to have had the coronavirus infection confirmed by a medical test. Since these participants may differ in their responses to other variables in the hypotheses, we have excluded them from the sample. None of hypotheses changed direction or effect size with the inclusion of these participants. As a result of excluding participants reporting to have been infected, hypotheses H4 and H5 related to the effects that perceived risk and compliance had on coronavirus infections will not be tested. The provided data and analysis scripts should make it possible for interested readers to rerun analyses with these participants reinstated.

## Materials and methods

### Participants

The data was collected through the "Norwegian Citizen Panel" (https://www.uib.no/en/citizen), which is an online platform for surveys of opinions of various societal matters in Norway. The participant panel had been surveyed 17 times previously, going back to 2013. The University of Bergen is responsible for running the panel, while the company Ideas2evidence

**Table 1. List of items used in analyses.**

| Variable | # | Item text (translated to English) | Mean (and SD) |
|---|---|---|---|
| Perceived risk | 15 | How high or low do you think the risk is that you will be infected by the coronavirus during 2020? | 2.91 (1.05) |
| Perceived risk | 48a | How big do you consider the risk that by 2020. . . an average adult will be infected by the coronavirus | 3.42 (0.99) |
| Perceived risk | 48b | How big do you consider the risk that by 2020. . . you will become seriously ill by the coronavirus | 2.16 (0.96) |
| Perceived risk | 48c | How big do you consider the risk that during 2020. . . your everyday life will change a lot due to the coronavirus | 3.58 (1.08) |
| Perceived risk | 40 | I worry that I will be infected by the coronavirus. | 2.74 (1.07) |
| Perceived risk | 41 | I worry that someone in my family will be infected by the coronavirus. | 3.74 (1.05) |
| Optimism | 20 | Would you say that most people in general are to be trusted, or do you think that one cannot be careful enough when dealing with others? [10-step item rescaled to 5-step] | 6.69 (1.06) |
| Optimism | 49 | It would be very serious for me if I got infected by the virus. [scores reversed] | 3.03 (1.12) |
| Optimism | 50 | I am optimistic that I will deal with the challenges that the corona outbreak will give me. | 3.93 (0.72) |
| Optimism | 74 | How confident do you feel that you will receive good treatment in the public health system if you become acutely and seriously ill? [7-step response for certainty, rescaled to 5-step] | 4.54 (0.94) |
| Exposure | 5 | Has anyone in your household or close family (parents, in-laws, children, siblings) been infected with the coronavirus? | 0.03 (0.17) |
| Exposure | 6 | Has anyone among your closest contacts been infected with coronavirus? | 0.09 (0.29) |
| Exposure | 9 | Have you been abroad in 2020? | 0.32 (0.47) |
| Exposure | 11 | Has anyone in your household or close family (parents, in-laws, children, siblings) been quarantined due to suspected coronavirus infection? | 0.31 (0.46) |
| Compliance | 45 | To what extent do you trust the health authorities's advice about the pandemic? | 4.19 (0.66) |
| Compliance | 44a | I do my best to follow the various advice from health authorities to limit the risk of infection (often washing hands, avoiding travel and situations with other people, keeping distance and avoiding touching things) | 4.66 (0.73) |
| Compliance | 44b | By following the infection control measures I will avoid getting sick | 4.06 (0.95) |
| Compliance | 44c | By following the infection control measures, I will avoid making others sick | 4.43 (0.79) |
| Information gathering | 46 | It is important to me that the information about the disease comes from a credible source. | 4.78 (0.57) |
| Information gathering | 47 | I believe that information about the coronavirus is deliberately kept concealed from us. [scores reversed] | 4.08 (0.96) |

Table 1 shows means and standard deviation for all items used in the current analyses. As descibed elsewhere in the Materials section, not all items described in the preregistration are included.

recruit participants, create the survey and document the data collection. Norwegian citizens above the age of 18 qualify to participate and the panel aims to be representative of the Norwegian population across a number of demographic variables.

The sample is mostly representative for the Norwegian population, but deviates somewhat in terms of age, education level and urban living. People in the age group 59 or older were overrepresented by 15%, while people between 18 and 29 years old were underrepresented by 12%. People with university level education were overrepresented by 29%, whereas people with elementary or no education were underrepresented by 19%. Furthermore, the three largest metropolitan areas in Norway were overrepresented by 4% in the sample. The deviations

are similar to that of prior data collections in the Norwegian Citizen Panel and are thus unlikely to be associated with the variables being measured in the current data collection (see documentation report for details: https://osf.io/uebq7/).

The survey was sent out to 15,409 eligible respondents (among those who had responded to any of the last three survey rounds). A higher-than-expected response rate resulted in N = 12,057 valid responses, compared to the 10,000 that was anticipated in the preregistration. The sample size was predetermined by the data collection agreement, and the researchers had no possibility for adjusting the data collection after examining parts of the data (i.e., no possibility for optional stopping).

There were three different versions of the survey, and the questions about the themes described in the current article were mainly included in one of the versions. This resulted in that actual sample size of N = 4,083 for the current analyses.

## Data collection

The Norwegian Citizen Panel recruited participants by random selection from the Norwegian Tax Administration registry. Invitations to the current survey were sent out to panel members by email on March 20, 2020 [42]. Most responses were received within the first days, but the survey remained open for responses until March 29. Those who had not opened or completed the survey received a reminder a few days later. Only completed surveys are included in the sample. The data collection period may be thought of as the "early phase" of the COVID-19 pandemic in Norway, as the government had imposed severe infection control measures on March 13. These measures included closing educational institutions and establishments in the service sector, discontinuing organized sports activities and other cultural events, and some restrictions on public transportation. Additionally, the government announced a number of recommendations such as keeping physical distance to others, working from home, avoiding travel, and hand sanitation advice.

The methodology report (https://osf.io/uebq7/), codebook (https://osf.io/xb6dk/) and analyses (https://osf.io/khbvz/) are available online. A previous publication [41] from the same dataset provides more details about the data collection, and provides descriptive statistics for the sample, figures showing the response distributions, and discusses some covariates.

## Materials and variables

The 20 items in the survey that were relevant for hypotheses H1, H2 and H3 were classified in accordance with the preregistration into five different variables. Variable indexes, order in the survey, and full item text are shown in Table 1 (the variable list is also available online at https://osf.io/x549e/). When not otherwise indicated, these were 5-step Likert-style items with statements about the pandemic to which the participant indicated their agreement from "completely disagree" (1) to "completely agree" (5).

**Perceived risk measure.** The preregistration listed six items related to "Perceived risk". Participants indicated how large they saw the probability that they themselves or others would be infected with the virus, would contract the disease and that that their everyday life would be changed by the pandemic. They also indicated their agreement to two additional statements about being concerned about being infected themselves or concerned about other being infected. The internal consistence between the six "Perceived risk" items was acceptable (Cronbach's alpha of 0.726).

Note that the survey included two yes/no questions about being unsure about having been infected and about wanting to be tested for infection, that the preregistration listed as being related to "Perceived risk". These questions were intended to estimate the extent of the

pandemic in Norway at that time, and were originally not planned as part of the current research project. These items are difficult to align with a clear theoretical construct of "Perceived risk" and will be excluded from the hypothesis testing.

To test H1, H2 and H3, an index was calculated as an arithmetic average of these six items, where a higher score indicates seeing the pandemic as constituting a larger risk. To test H1c an index for degree of optimism bias was calculated as the agreement score for "risk for self" subtracted from the score for "risk for others".

**Optimism in managing the pandemic measure.** There were four items related to optimism. An item asked whether you think that people in general are to be trusted. This item has previously been used as a measure of "societal trust" [40, 43]. It has been argued that social trust is part of a broader personality complex that include optimism, a belief in co-operation, and confidence [44]. For our research interests, societal trust may thus be relevant for how individuals expect to fare in the pandemic. In addition, there were three items adapted to the current pandemic setting that asked whether you believe that becoming infected would not be serious for you, thinking that you would be able to handle the challenges of the pandemic, and that you would receive good treatment if become sick. The two items that had been measured with seven-step and ten-step scales (in order to compare with measures at earlier data collections) were rescaled to a five-step scale (see Table 1). An average of the four items was taken to express "Optimism in managing the pandemic" to test H1a-d. The internal consistency between the four "optimism" items was acceptable (Cronbach alpha of 0.73).

The preregistration also described an item about having received sufficient information about the pandemic as being related to "Optimism": However, due to technical issues this item was not actually recorded for the sample being analysed here.

**Exposure measure.** Four items were used as a measure of potential exposure to infection. The items were related to whether anyone in your household or your closest contacts had been infected or had been quarantined, or whether you had been abroad so far this year (a significant portion of those infected in Norway at the time of measurement had been infected or secondary contacts of people who had been abroad). Responses were scored 1 for "yes" and 0 for "no", and an average of these four items was used as an index to indicate the variable "Exposure" to test hypothesis H1b. The four items had low internal consistency (coefficient omega of 0.264) indicating that they should not be considered to represent a unitary theoretical concept. This is acceptable for our purposes, as they represent being exposed in various discreet settings, that can be expected to vary independently.

**Compliance measure.** Four items measured agreement to statements about trusting and intending to follow the infection control measures. One item asked about trusting the advice about the infection control measures, two about being motivated for following the measures, and one about intending to follow the measures. An average of these four items was used as an index for "Compliance" to test H2. The internal consistency between the four "compliance" items was acceptable (Cronbach alpha of 0.71).

**Information gathering measure.** Respondents indicated how important they think it is that information about the pandemic comes from a trustworthy source, and whether they trust the available information about the pandemic. The index for "Information gathering" consisted of two potentially orthogonal items about caution and suspicion in information gathering, which had low internal consistency (Spearman $r$ = .2). This suggest that the two items should be analysed separately, and were therefore followed-up with independent tests for each outcome item. Further, trusting information about the pandemic was found to be correlated with believing that it was important that information comes from a trustworthy source.

### Ethics statement

All ethical aspects of the data collection and data storage in the Norwegian Citizen Panel are approved by the Norwegian Centre for Research Data (reference number: 118868). A written informed consent form was obtained from all panel members ahead of the data collection.

## Results

### Response distributions

Table 1 shows means and standard variation for all items used in the current analyses. The average score on the "Perceived risk" index (M = 3.09, SD = .68) and the items constituting it showed that most participants chose responses comparable to "medium risk", but with standard deviations extending to the response option above ("somewhat high risk") and below ("somewhat low risk"). The "four "Optimism" items were scored on a scale from 1–5, and the average index approached high optimism (M = 3.78, SD = .49). As shown in Table 1, participants were very optimistic that they would receive good medical treatment if sick, and highly optimistic that they would handle the challenges of the pandemic. They were somewhat less optimistic in general, and were moderately optimistic about how serious it would be for them to be infected. Average scores to the "Exposure" index (M = .19, SD = .2, on scales with 1 for Yes and 0 for No) showed that participants mostly answered in the negative. In more detail, relatively few had household members (4%) or close contacts (10%) that they assumed had been infected at that time, while more of them had been abroad in the last three months (34%) or had close contacts that been quarantined (32%). There was high agreement about the compliance items ("Compliance" index M = 4.33, SD = .58), indicating that most participants trusted the infection control advice, intended to follow it, and believed in its efficiency. An average of the two "Information gathering" items (M = .4.45, SD = 0.6) showed this to be emphasized by most participants. As shown in Table 1, most agreed that it was important that information about the disease came from a credible source, and most trusted the official pandemic information (but note the larger variability for this item). Some of these response distributions have been discussed in more detail and illustrated in a previous article [41].

### Effects of optimism

**Optimism and perceived risk.** The H1a hypothesis anticipated that optimistic thinking about the pandemic would be associated with lower levels of perceived risk. A regression of "Optimism" on "Perceived risk" ($F(1, 3950) = 440.53$, $p < .001$, adj. $r^2 = 0.1$) explains a statistically significant and small proportion of variance in the predicted negative direction. This indicates that more optimistic people see the risk as lower, as shown in Fig 1.

Three of the items in the optimism index were related to being optimistic about the outcomes of the pandemic. In order to test for the effect of context-free dispositional optimism, the H1a test was followed-up with an unregistered regression of a single-item indicator of optimism (*"most people are to be trusted"*) on perceived risk. This follow-up test still found a significant negative effect, but of a smaller magnitude $F(1, 3852) = 122.18$, $p < .001$, adj. $r^2 = 0.03$).

**Optimism, exposure and perceived risk.** As an extension of H1a, the hypothesis H1b anticipated that for a given level of exposure to potential contagion, more optimistic people may experience the risk as lower. A multiple regression ($F(3, 3948) = 159.88$, $p < .001$, adj. $r^2 = 0.11$) showed that "Optimism" had a significant effect on "Perceived risk" $t(3948) = -18.82$, $p < .001$, reflecting the same effect as in H1a), whereas "Exposure" did not ($p = .43$), and there was no interaction effect ($p = .87$).

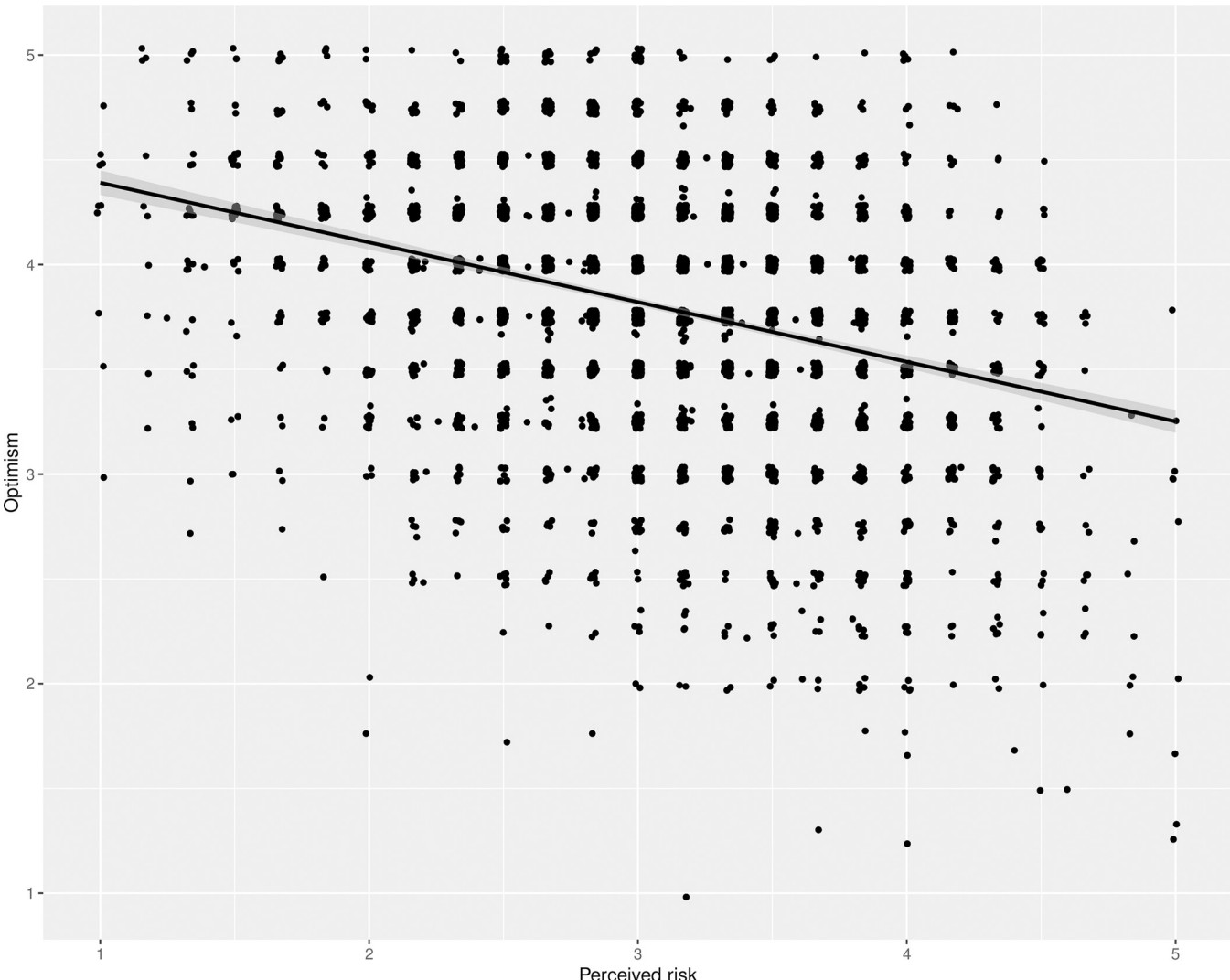

**Fig 1. Regression of optimism on perceived risk.** The figure shows the regression line with 95% confidence intervals in grey, corresponding to a test of hypothesis H1a, on top of a jittered scatterplot of the responses.

**Optimism and optimism bias.** As can be seen in Table 1, the "risk for others" was rated as higher than the "risk for self". A t-test ($t(7612.89) = 18.73$, $p < .001$; $d = 0.42$) showed a significant medium sized difference. The H1c hypothesis anticipated that this optimism bias would have a positive association with optimism. A simple regression found a positive association that explained a small proportion of the variance ($F(1, 3890) = 409.98$, $p < .001$, adj. $r^2 = 0.1$). This indicates that optimistic people to a larger extent overestimated their own safety relative to that of others.

**Optimism and compliance.** The H1d hypothesis anticipated that higher levels of optimism about handling the pandemic would be associated with lower compliance to the infection control measures. This was tested with a simple regression with "Optimism" as a predictor and "Compliance" as an outcome. This test showed a very small effect in the opposite direction of what was predicted ($F(1, 3931) = 66.23$, $p < .001$, adj. $r^2 = 0.02$). This indicates that more optimistic participants were more likely to say that they would comply with the measures, as shown in Fig 2. An explorative regression model of the optimism items on the

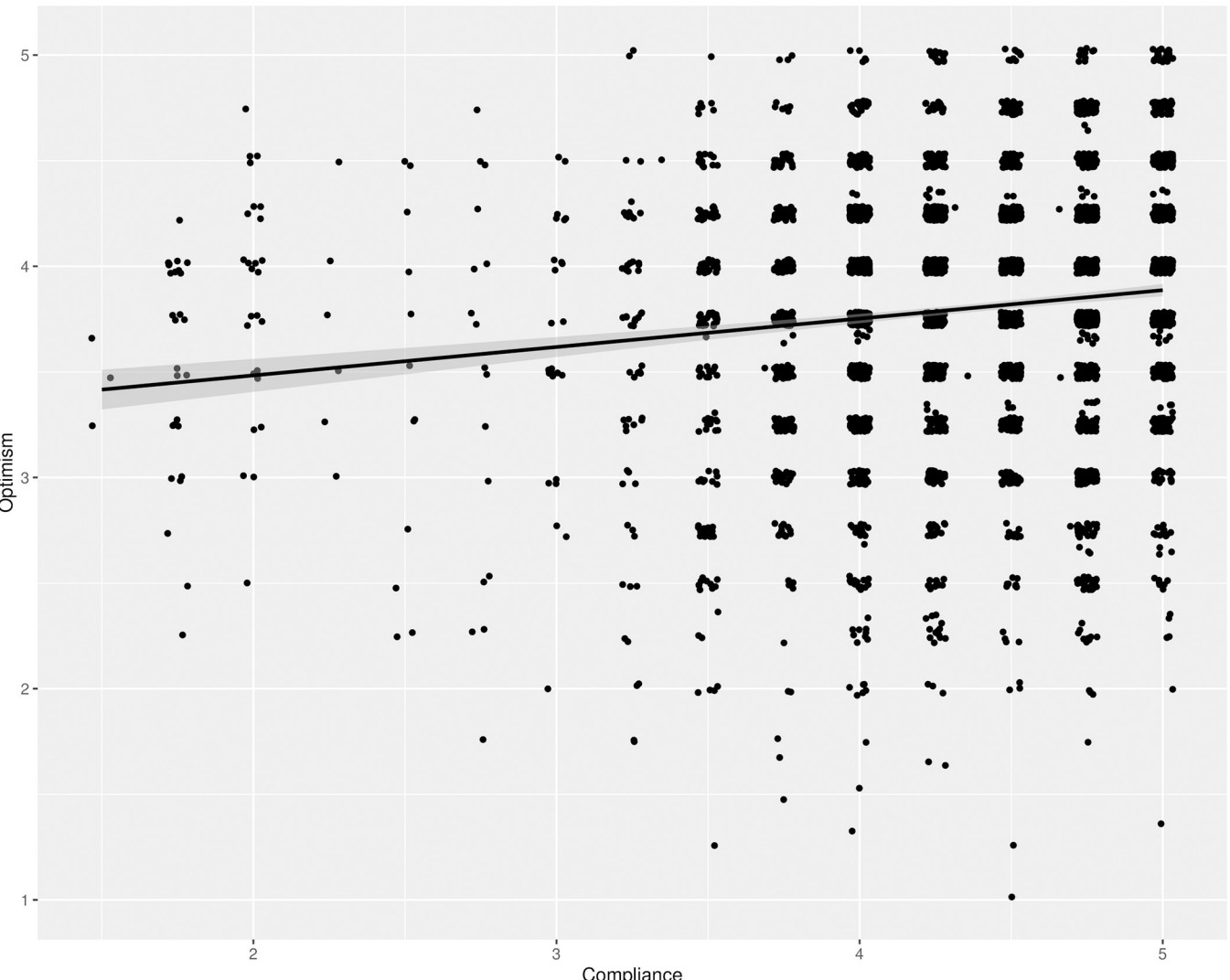

**Fig 2. The regression of optimism on compliance.** The figure shows the regression line with 95% confidence intervals in grey, corresponding to a test of hypothesis H1d, on top of a jittered scatterplot of the responses.

compliance index showed that the item about how serious an infection would be for them loaded negatively on compliance. Without this item the remaining optimism items had a somewhat larger positive effect on compliance ($F(1, 3920) = 136.41$, $p < .001$, adj. $r^2 = 0.03$).

### Effects of perceived risk

**Perceived risk and compliance.** The H2 hypothesis anticipated that those who see the risks from the pandemic to be higher would be more likely to comply with the infection control measures. A regression showed a significant but negligible effect in the opposite direction, that higher "Perceived risk" was associated with lower "Compliance" ($F(1, 3931) = 5.63$, $p = .018$, $r^2 = .001$). This indicates that those that see a larger pandemic risk are not more likely to comply with the measures. This finding justifies further exploratory analyses of the relationship between risk and compliance. Replacing the risk index with the item about risk for being infected showed a very small negative effect on compliance($F(1, 3929) = 56.38$, $p < .001$, $r^2 =$

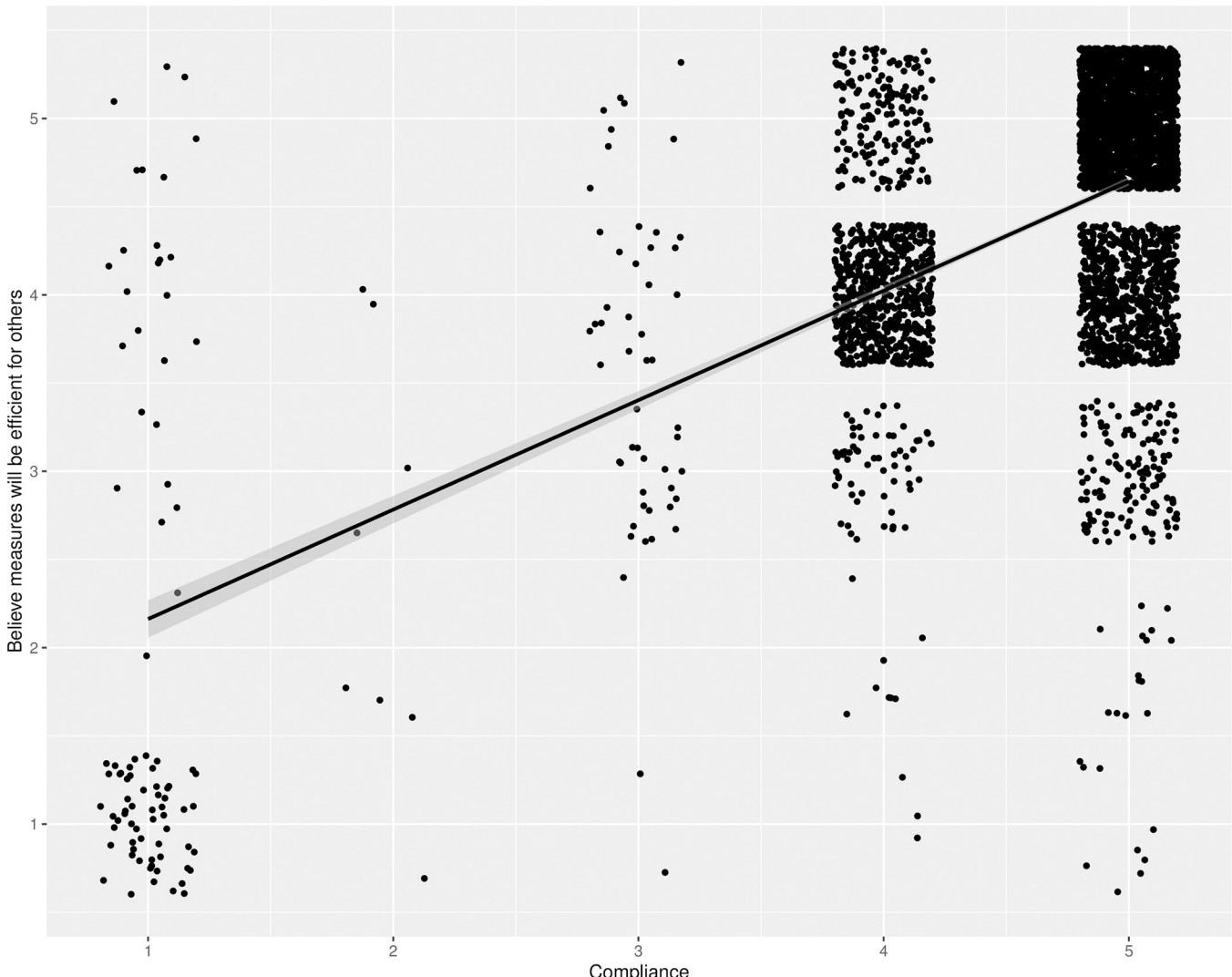

**Fig 3. Regression on compliance of believing that measures will be efficient in protecting others.** The figure shows the regression line with 95% confidence intervals in grey, corresponding to a test of hypothesis H2, on top of a jittered scatterplot of the responses.

.01). The item about perceived risk for people in general had a negligible positive effect ($F(1, 3888) = 4.81$, $p = .03$, $r^2 = .001$). There was thus only weak support for hypothesis H2, that seeing the pandemic as a threat would correspond to following the control measures, and only when asking about risk for the general population, but not for risk for oneself.

**Prosocial motivation and compliance.** Since the above H2 analysis indicated that there may have been small but opposite effects of personal and general perceived risk, we wanted to examine the relationships between the items constituting the overall compliance index. As an exploratory analysis, we tested whether compliance was associated with seeing the infection control measure as being effective in protecting others. There was a substantial effect of believing that by following the measures one would avoid infecting others($F(1, 3918) = 1855.15$, $p < .001$, $r^2 = .32$), as shown in Fig 3. There was also a significant but smaller effect on compliance of believing that by following the measures one would avoid becoming infected oneself ($F(1, 3921) = 814.65$, $p < .001$, $r^2 = .17$). Adding the "keeping you safe" to a regression of "keeping others safe" did not increase the total explained variance ($F(1, 3916) = 947.68$, $p < .001$, $r^2 =$

.33). This indicates that seeing the measures as being effective in protecting others was associated with following the measures, more so than seeing the measures as being effective in protecting yourself.

Note that our measure of compliance included questions about attitudes and motivation in addition to intention to comply. In order to test the effect on intention to comply alone, we also ran the tests of H1d and H2a on the single item for compliance, instead of the index across all compliance items. These analyses can be found as supplemental online materials (https://osf.io/khbvz/).

## Effects on information gathering

Hypothesis H3 predicted that "Perceived risk" would be positively associated with "Information gathering". Testing this found a very small negative effect of "Perceived risk" on an index of information gathering ($F(1, 3930) = 18.42$, $p < .001$, adj. $r^2 = .004$). The index consisted of two items that do not necessarily indicate the same underlying theoretical construct (i.e., they are potentially orthogonal). The preregistered analysis was therefore followed-up with a regression where both single-items were added as regressions. This analysis found a very small positive effect of perceived risk on the item about placing importance of credible sources (adj. $r^2 = .004$), and a somewhat larger negative effect on the item about trusting the official information about the pandemic (adj. $r^2 = .02$). See this and other analyses in supplementary materials online (https://osf.io/khbvz/).

## Discussion

### Summary of results

We performed preregistered analyses on data from a representative survey that was collected in the first weeks after the COVID-19 lock-down was instituted in Norway. Both optimism and perceived risk had small and unclear associations with compliance, while prosocial motivation had a clear association with compliance.

**Optimism and risk.** Our hypotheses assumed that optimistic people would see the pandemic risk as smaller (H1a). This was supported with a small effect in the predicted direction, but it did not interact with the degree of infection exposure (H1b). There was an overall optimism bias in the sense of seeing themselves as safer than others, and a small effect that more optimistic people had a larger optimistic bias (H1c). We expected that optimistic participants would comply less (H1d), but we found a small effect in the opposite direction.

Note that while the concept of risk is typically considered as a factor of both probability and consequence, the current measure of risk is mainly focused on the probability aspect of risk. This was due to assuming that the consequences of the pandemic at that time was unknown and uniform among the participants.

**Risk and compliance.** The hypothesis (H2) that compliance would be higher for those that saw the risk as larger was not supported. An explorative analysis of items related to motivation found that believing that compliance would be effective in protecting others had a moderate association with compliance. A similar test of self-interested motivation showed a smaller effect.

**Information gathering.** We assumed that those that saw the risk as larger would be more careful in their information gathering (H3). However, there was only a weak negative association between perceived risk and an index of the two items for information gathering. Examining this in more detail showed that risk indeed had a week effect on placing importance on credible sources, but a weak negative effect on trusting official information.

## Relationships between optimism, risk and compliance

**Optimism and perceived risk.**   Dispositional optimism, as the tendency to expect positive outcomes across a number of situations [14], may be expected to be associated with positive outcome expectations for specific events such as an ongoing pandemic. Consistent with this, we found dispositional optimism to be associated with seeing the risk of being infected as lower. Given the conceptual similarity between optimism and risk, it is perhaps surprising that there was only a small effect size for this association. This may indicate that the perceived risk of infection is impacted by a number of factors besides dispositional optimism, such as perceived vulnerability and exposure.

Optimism bias [22] may lead people to expect the pandemic to be worse for others than for themselves. This was supported in the current survey, as we found that our participants evaluated it to be less likely that they were infected than that the average person was infected. People may give lower risk estimates for themselves if they feel that they can control the risk by taking actions such as following the infection control measures or limiting their exposure. This corresponds to previous research that has indicated that people see uncontrollable risks as more threatening than risks of comparable magnitudes that they can control [45, 46]. This may be associated with issues of locus of control [47] and self-efficacy [48].

We hypothesized that the magnitude of the optimism bias would be associated with the degree of optimism. This was supported, in that optimistic people to a larger extent reported that the risk was larger for others than it was for themself. It has previously been argued that optimism bias may be a challenge for managing pandemics, at it leads people to underestimate the risks of being infected [49, 50]. If so, our results indicate that such challenges may be particularly pronounced for people that are more optimistic in general and have optimistic evaluation of the pandemic. However, we argue below that in some pandemic situations prosocial motivation may be more important for compliance than perceived risk. It is thus possible that despite an optimistic bias making people see themselves as safe, they may nevertheless be motivated to comply in order to protect others that they see as being at risk.

It should be mentioned that the current study did not use an established scale for dispositional optimism, such as the "Revised Life Orientation Test" (LOT-R). The single-item measure we used about whether people find most others to be trustworthy has frequently been used as measure of "societal trust" [40, 43], and has been argued to be relevant for optimism [44]. Later studies in the project will compare this operationalisation of pandemic optimism to validated measures of dispositional optimism.

**Optimism and compliance.**   A general finding in research on past pandemics has been that feeling threatened by a pandemic leads to self-interested motivation to comply with infection control measures [1, 11]. Optimism may be a possible moderating variable for this relationship. For a given level of pandemic threat, those with a more positive outlook on life in general or on the pandemic in particular may feel that their own outcomes will be better than those who are less optimistic. Higher optimism could lead to a sense of complacency, indifference or nonchalance towards pandemic risks. This could have made more optimistic people feel that they would be safe and manage quite well regardless of whether they follow the measures or not [49]. If self-interested motivation to prevent negative consequences is crucial for compliance with infection control measures, optimism could thus reduce compliance.

However, our results indicated the inverse relationship, that optimism had a small *positive* association with following the infection control measures. This initially surprising finding indicates that a more optimistic mindset could contribute positively to compliance through providing motivation and drive to follow the measures [15]. It could be that pessimism, in the sense of not expecting the pandemic situation would improve over time or that oneself to

manage its challenges, could decreased the motivation to comply with the measures. To comply with the measures involves changing and adapting our daily routines and activities, often to the detriment of quality of life. These costs of compliance combine with lacking certainty about how effective the measures will be effective in protecting oneself and others from a novel and unknown pandemic. Optimism may also be related to believing that the measures will be effective, which may drive the positive association between optimism and compliance. It should be noted that while past research tends to support an optimism bias, the assumed positive association between optimism and health protective behaviours has been more difficult to demonstrate [see e.g., 51]. Similar to our study, previous studies have also shown optimism to have positive associations with compliance [e.g., 30].

Even at the early stage of the pandemic when the current data was collected, people already had some experience with the infection control measures and may have expected the measures to continue for some time. Some participants may have suspected that they would have difficulties in completely abiding with all the measures over time. Thus, an optimistic and positive mindset may have provided the necessary enthusiasm and perseverance to anticipate being able to comply over time. This view of optimism may be related to the concept of self-efficacy [19], which has been shown to be necessary for the adaptation of protective behaviour [21]. More specifically, optimism has been shown to predict more active or problem-focused coping [52]. Dispositional optimism has been associated with less avoidant coping after HIV testing, and AIDS-specific optimism was associated with motivation for health promoting behaviour [53]. Previous COVID-19 research [49] has indicated that optimism has been related to the perceived level of control over pandemic risks. Another recent study on a representative Norwegian sample [54] found optimism bias to not be related to willingness to vaccinate. Overall, these findings indicate a more complex relationship between optimism and compliance than we initially assumed. Different mechanisms may drive both positive and negative effects of optimism, and in the current data the balance tips slightly in favour of a positive effect.

**Perceived risk and compliance.**   As discussed above, we did not find the expected association between optimism and reduced compliance. This expectation was based on an assumption that compliance is based on self-interest, and that the self-interest would be smaller for more optimistic people that believed they would manage well regardless of their compliance [see similar arguments in 12, 13]. This assumption has been supported by findings such as that in the H1N1 pandemic, precautionary behaviour was associated with seeing the disease as severe and the risk of being infected as high [55]. As indirect support, it has been found that at-risk groups are more motivated to be vaccinated [1]. However, more recent studies have indicated that self-interested concern for your own safety may not be as crucial motivation for compliance as previously assumed [56, 57].

In line with more recent research, our preregistered analysis did not support the expected positive association between risk and compliance. We found no indication that feeling threatened by the pandemic motivated people to follow the infection control measures. We should note that the data was collected at a relatively early stage of the pandemic, during sharp increases in infection rates, and the far majority of the population was aware of the risks and motivated to comply. Perceived risk and compliance may thus have been higher and had less variability in our population than in the populations of past studies. The observed relationship may also have been influenced by factors related to the Norwegian culture, public health program and pandemic response at the time. This should be kept in mind when reflecting on why the relationship between risk and compliance was different in the current study from other studies in the literature.

Explorative analyses found a very small effect that the perceived infection risk for the general public was associated with compliance. Since the positive effect was for risk for others, but

not for oneself, this also goes against the assumption that self-interested motivation is crucial for compliance. This may support the argument (see below) that prosocial motivations are more important for compliance than self-interest.

**Motivation and compliance.** Compliance with infection control measures that required change of everyday behaviour is likely to be determined by a variety of factors such as habits, social norms, structural constraints and random factors. To the extent that people make rational choices about compliance, it is determined by the goals they set and their motivation to follow those goals. In line with the theory of planned behaviour [21], one would expect that the positive attitude, the behavioural intention and the social motivation would predict actual behaviour. The theory has previously been applied to a wide range of decision-making, such as the prediction of vaccine uptake during the H1N1 pandemic [58].

As discussed in the preceding section, we found no indication that feeling personally at risk increased compliance to infection control measures. This led to an explorative analysis that showed that those who believed that the measures were effective in protecting others from infection complied to a larger extent. This association accounted for a substantial portion of the variation in compliance. In addition, seeing the measures as effective in protecting oneself was also positively associated with compliance, but explained less of the variation. It could be argued that people who mostly think of compliance in terms of protecting others (rather than protecting themselves) have more prosocial, rather than self-interested motivation for complying. In this sense our results indicate that prosocial motivation may be one of the crucial factors to explain variation in compliance to infection control measures.

To have prosocial motivation indicates that people are moved to act based on concern for others, empathy, in the sense of seeing the needs and perspectives of others, and altruism, in the sense of wanting to help others without aiming for personal gain. Previous research on compliance has suggested that people chose to engage in protective behaviour based on a conscious weighing of costs and benefits [59]. While benefits of compliance (e.g., reducing infection rates) are shared by both the individual and community, costs are often incurred by the person engaging in the protective behaviour (e.g., the negative sides of self-isolating or wearing a facemask). Compliance may thus be viewed as at least a partly a prosocial act. A meta-analysis [60] supported the current results by showing that compliance was not associated with perceived risk alone, but that an interaction between feeling threatened and believing in the efficacy of the infection control measures produced the greatest adaptation of protective behaviour. A recent study showed that to highlight the public benefits increased behaviour that was precautionary against COVID-19 infection [26]. This indicates the importance of prosocial motivation for compliance, comparable to the current study.

It should be noted that our analysis of different types of motivation was exploratory, as opposed to the confirmatory tests of preregistered hypotheses discussed above. Relatedly, the questions about trusting the infection control measures' efficiency were not designed to differentiate between different types of motivation, and may not be the optimal approach for such an issue. This should be taken into account when interpreting the effect. The identified effect and the potential mechanism behind it will be further tested as preregistered hypotheses in future survey rounds in the current project.

**Perceived risk and information gathering.** Our descriptive results showed that a majority of the population emphasized both that information about the pandemic should come from credible sources and they trusted that information was not being kept away from them.

We wanted to test whether information gathering varied with perceived risk. Our questions about placing importance on credible sources and on trusting official information turned out to have opposite associations with perceived risk. We found that the majority view of emphasizing credible sources was somewhat more frequent among those that saw the pandemic

threat to be serious. Although very weak, such an effect may indicate that taking the risk seriously is related to also thinking critically about which information sources that can be trusted. However, the association could also be due to a third factor (such as knowledge about infection mechanisms) that causes both higher perceived risk and emphasizing credible sources. Note that people's ideas of trustworthiness may differ, and what some consider a credible source may be considered as conspiratorial by official standards.

Rather few in our sample believed that information about the pandemic was being hidden or concealed, but this view was somewhat more common among those who saw the risk as high. We should note that this question is somewhat open to interpretation. It could refer to what may be considered healthy scepticism, such as believing that some governments were not forthcoming in sharing information about the origin or spread of the pandemic [61], or that local infection rates are not made available to them as quickly and openly as they would have liked. The phrasing of the item could also be interpreted as being in support of conspiracy theories, such as believing that governments or pharmaceutical companies were concealing their role in manufacturing or propagating the pandemic [62]. Our results show that those who trust the official information see the risk as somewhat lower. If having misgivings about the availability of information was to indicate believing in a conspiracy where the pandemic risk was being downplayed, one would expect the opposite association. The current study is not suited to evaluate conspiratorial thinking about the COVID-19 virus. Other studies that have explicitly focused on this issue have indicated that conspiratorial thinking may be associated with seeing the risk of the pandemic as low, and also associated with reduced compliance with infection control measures [63].

Knowledge about the public's information gathering can be applied to planning public health information campaigns. The concept of information seeking [64] has only been partially developed, and may be considered as both a state and a trait characteristic [65]. The "planned risk information seeking model" [64] has linked the gathering of health information to the theory of planned behaviour [21]. The current results indicate that most Norwegians at the time had information gathering approaches that were conducive for complying with the official infection control measures. Variation in perceived risk did not meaningfully influence the information gathering approaches.

## Limitations, implications and future research

**Limitations of the study design.** There are inherent weaknesses in cross-sectional survey data collections. It should be noted that the associations discussed above are correlational and should not be taken to imply causality. Although it may seem reasonable that dispositions predict intentions, there may also be third factors behind the associations [e.g., individual differences in social-desirability bias, 66, shared method variance, 67, or response mode when answering the survey, 68].

Note that the measures in the survey were constructed ad hoc to be suitable for the pandemic situation in Norway at the time. We should therefore seek to follow up the current findings using psychometrically validated measures. It should also be pointed out that the measure of compliance is measured as behavioural intentions, and there may be a number of reasons why actual behaviour may be different from a stated intention. As noted above, the operationalization of the concept of information gathering is somewhat open to interpretation as to what the participant considers to be a credible source.

**Implications.** The current study examined motivational factors and intentions from the beginning of the pandemic in Norway. Most previous research has indicated that perceived risk is associated with compliance[1, 11]. If such a relationship holds true, one may want to

caution the public against having an overly optimistic view of how they will manage the risks of the pandemic. In the current study, perceived risk accounted for little of the intentions to comply with infection control measures. Instead, the current results indicate that motivation type may play a more important role for compliance. This may indicate that there are more complex relationships between probability, consequences and compliance than what has previously been assumed. The role of motivational factors, such as believing in the efficiency of the measures for keeping themselves and others safe have been less explored in previous research on protective medical behaviour.

The current study may have some implications for public health communication during epidemics. Emphasizing a threat in order to increase compliance must be balanced against the negative effects the approach has for individuals and for the society. Living for prolonged periods in a state of fear may lead to short-term negative emotions (stress, anxiety and concern), longer-term mental health effects of living in a state, and may detriment inter-group relationships and democratic values. In settings like that currently studied, where the majority of the population already has a realistic impression of the pandemic risk [69] there appears to be little to gain in emphasizing the threat in order to increase compliance. Given the potential negative effects of prolonged states of perceived high risk, the current results discourage such public health approaches. Convincing the public about the efficacy of infection control measures and the public benefit of compliance may be a more advantageous approach. Further, emphasizing public benefits may have advantages of being effective also in situations where the receiver is not personally threatened (e.g., for a young person who believes the pandemic is only a threat for older people).

The current study was done in Norway in the early phases of a widely discussed pandemic, where the infection control measures can be assumed to be well known and most of the population considered the risk to be fairly high. This context is different from that of most past studies on determinants of compliance to infection control measures. This difference in context may have impacted the results, and one should be careful in generalizing the current results to vastly different settings.

**Future research.** The results reported here was from the first of several survey rounds of data collection in a larger project (see more information at https://www.uib.no/pandrisk). Future surveys will follow-up on some of the trends reported here, and will measure aspects of risk, compliance, motivation, optimism. More sophisticated preregistered analyses will test the relationships and relative contributions from these factors on compliance.

## Acknowledgments

### Author note

Thanks to members of the PANDRISK research project for discussing the survey design and interpretation. Thanks to all the participants from the NCP that took time to respond to the survey.

The analyses in the current manuscript were publicly preregistered in advance of accessing the data (https://osf.io/umgnr). A previous publication [41] has reported descriptive statistics from the same dataset as the current manuscript.

Data was collected as part of the COVID-19 extraordinary data collection of the "Norwegian Citizen Panel", collected in March 2020 (Ivarsflaten et al., 2020).

## Author Contributions

**Conceptualization:** Bjørn Sætrevik.

**Data curation:** Bjørn Sætrevik, Sebastian B. Bjørkheim.

**Formal analysis:** Bjørn Sætrevik, Sebastian B. Bjørkheim.

**Funding acquisition:** Bjørn Sætrevik.

**Investigation:** Bjørn Sætrevik.

**Methodology:** Bjørn Sætrevik.

**Project administration:** Bjørn Sætrevik.

**Resources:** Bjørn Sætrevik, Sebastian B. Bjørkheim.

**Writing – original draft:** Bjørn Sætrevik, Sebastian B. Bjørkheim.

**Writing – review & editing:** Bjørn Sætrevik, Sebastian B. Bjørkheim.

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
