## [Decision Letter · Decision Letter 0]

26 May 2022

PONE-D-22-08667Motivational factors were more important than perceived risk or optimism for compliance to infection control measures in the early stage of the COVID-19 pandemicPLOS ONE

Dear Dr. Sætrevik,

Thank you for submitting your manuscript to PLOS ONE. After careful consideration, we feel that it has merit but does not fully meet PLOS ONE’s publication criteria as it currently stands. Therefore, we invite you to submit a revised version of the manuscript that addresses the points raised during the review process.

We look forward to receiving your revised manuscript.

Kind regards,

Ali Safaa Sadiq

Academic Editor

PLOS ONE

Journal Requirements:

Reviewers' comments:

Reviewer's Responses to Questions

**Comments to the Author**

1. Is the manuscript technically sound, and do the data support the conclusions?

Reviewer #1: Yes

Reviewer #2: Yes

2. Has the statistical analysis been performed appropriately and rigorously? 

Reviewer #1: Yes

Reviewer #2: Yes

3. Have the authors made all data underlying the findings in their manuscript fully available?

Reviewer #1: No

Reviewer #2: Yes

4. Is the manuscript presented in an intelligible fashion and written in standard English?

Reviewer #1: Yes

Reviewer #2: Yes

5. Review Comments to the Author

Reviewer #1: The relationship between optimism and compliance to infection control measures has been investigated previously by at least two studies (Hartmann, M., & Müller, P. (2022). Acceptance and Adherence to COVID-19 preventive measures are shaped predominantly by conspiracy beliefs, mistrust in science and fear - A comparison of more than 20 psychological variables. Psychological Reports. doi: 10.1177/00332941211073656; Jovančević, A., & Milićević, N. (2020). Optimism-pessimism, conspiracy theories and general trust as factors contributing to COVID-19 related behavior – A cross-cultural study. Personality and Individual Differences, 167, 110216. https://doi.org/10.1016/j.paid.2020.110216). The results from these (and possibly other) previous studies should be integrated in the introduction and discussion of results.

On p. 4, the authors argue that self-interested concern for your own well-being is a central motivation for compliance. This is certainly true, but what about the motivation to care for others, e.g., because many friens belong to the risk group, or if a person has a pronounced (pro)social behaviour? Also the big 5 trait agreeableness may play a role (Hartmann & Müller, 2022). Some of this is mentioned on p. 6, this could be integrated a bit better, also to avoid some redundancies in 1.1.2 and 1.1.3.

Optimism is a complex issue. The authors mention that optimism is likely to be associated with lower risk perception and lower compliance to preventive measures. However, also the attitude of whether the measures or compliance will be effective determine actual compliance. The authors well addressed these different possible facets of optimism. For this study, I was wondering whether it would not have been of importance to differentiate optimism also for their measurements (e.g., general optimism that things come out well for you, optimism of dealing with infection, and optimism of the effect of compliance to the preventive measures. Why did the author not measure these different facets?

1.4.2 the hypothesis regarding information gathering is difficult to understand when the reader does not know how “information gathering” is operationalized. For example, “only trusting sources they find credible and trusting the official information” is ambiguous. Some may find sources of fake news and conspiracies very credible, then a reverse association with perceived risk may be expected. Please clarify at this point whether the measurement focuses on official information gathering.

p. 9 “the reported symptoms make it less likely that they had in fact been infected by the coronavirus”. This is a rather speculative statement. There are many infected people that “only” had such described symptoms without high fever. The author shuld at least report some references to support such a statement.

Analysis.

Given that the subsample of participants with confirmed infections was too low for separate analysis, they should probably be excluded from the analysis. Also, the people without confirmed infections might differ in terms of motivation. This should be captured by a covariate “infected”, or alternatively also this subsample could be removed from the analysis.

I was surprised that the compliance measure included items about trusting the infection control measure. Such items rather capture attitude and not directly compliance in terms of actual adherence to measurements. The differentiation between attitude and behaviour was discussed in the introduction but seem to be ignored for the study.

p. 15 “most trusted the pandemic information” -> the “official” information is meant here, right?

Did the author test for possible covariates for the regression analyses? For example, compliance might be higher for older people (or particularly for people in the risk group). Was there any health-related information about the participants as to whether they belong to the risk group (preconditions)? These people possibly have a higher compliance that may be independent from optimism etc. Or do you think this is sufficiently captured by higher perceived risk? Maybe this issue should be discussed somewhere more explicitly.

In addition to reporting F-statistics, the authors could provide a summary graph of their hypothesis where the relevant regression estimates and their 95%Cis are depicted.

Reviewer #2: The authors report data collected during the early stages of the COVID-19 pandemic from a large Norwegian sample. Overall, I find the study well conducted. In particular, hypotheses were pre-registered and results are clearly reported and carefully discussed in the manuscript. To make it brief, I am very much in favor of seeing this manuscript published. I only have three comments:

1. Please show figures depicting the main regressions (or correlations). I would be very much interested in seeing the distributions of the measured variables. In some cases, there might be floor or ceiling effects (or heavily skewed distributions) that would affect the interpretation.

2. What are the relationships between the questions in one category, i.e., between the items grouped into the five variables in Table 1? Cronbach’s alpha would be a standard measure. For some items, the authors discuss specific effects but it would be generally important to see if the categorization of these items is warranted (based on the obtained empirical data).

3. The text is very clearly written with one exception. The term optimism is used in different circumstances. Sometimes it is used as a short form for “optimism regarding the risks of COVID-19.” Sometimes it is used as a more general measure. It is also a bit confusing in some cases how “optimism” is different from “optimism bias.” The authors could simply provide a table that lists the different usages of these terms and then stick to these throughout the manuscript (including the title and the abstract).

6. PLOS authors have the option to publish the peer review history of their article (what does this mean?). If published, this will include your full peer review and any attached files.

Reviewer #1: No

Reviewer #2: No

---

## [Author Response · Author response to Decision Letter 0]

12 Aug 2022

Dear editor and reviewers,

We are very grateful to the editor and the two anonymous reviewers for taking time to consider our manuscript. The reviewers made a number of comments and suggestions, all of which we found to be well-considered and meaningful. We have thoroughly revised all sections of the manuscript based on this review. How we have responded to each of the comments are detailed in the attached file, citing the revised manuscript text and their respective line numbers. Note that line numbers refer to the version of the manuscript where changes are not marked. 

Since one of the comments suggested to adjust our sample, we have recalculated our analyses, and updated the numbers reported in the manuscript, as well as updating the files for dataset, analysis scripts, and analysis output on OSF. Note that adjusting sample did not substantially change the result of the analyses.

We hope that the editor and reviewers agree that these revisions have improved the clarity and the contribution of the manuscript.

Best wishes on behalf of both authors,

Bjørn Sætrevik

---

## [Decision Letter · Decision Letter 1]

5 Sep 2022

Motivational factors were more important than perceived risk or optimism for compliance to infection control measures in the early stage of the COVID-19 pandemic

PONE-D-22-08667R1

Dear Dr. Sætrevik,

We’re pleased to inform you that your manuscript has been judged scientifically suitable for publication and will be formally accepted for publication once it meets all outstanding technical requirements.

Kind regards,

Ali Safaa Sadiq

Academic Editor

PLOS ONE

Additional Editor Comments (optional):

Authors could manage to address all the given comments by the reviewers, hence, I would like to recommend their manuscript for the possible publication.

Reviewers' comments:

Reviewer's Responses to Questions

**Comments to the Author**

1. If the authors have adequately addressed your comments raised in a previous round of review and you feel that this manuscript is now acceptable for publication, you may indicate that here to bypass the “Comments to the Author” section, enter your conflict of interest statement in the “Confidential to Editor” section, and submit your "Accept" recommendation.

Reviewer #1: All comments have been addressed

Reviewer #2: All comments have been addressed

2. Is the manuscript technically sound, and do the data support the conclusions?

Reviewer #1: Yes

Reviewer #2: Yes

3. Has the statistical analysis been performed appropriately and rigorously? 

Reviewer #1: Yes

Reviewer #2: Yes

4. Have the authors made all data underlying the findings in their manuscript fully available?

Reviewer #1: Yes

Reviewer #2: Yes

5. Is the manuscript presented in an intelligible fashion and written in standard English?

Reviewer #1: Yes

Reviewer #2: Yes

6. Review Comments to the Author

Reviewer #1: (No Response)

Reviewer #2: (No Response)

7. PLOS authors have the option to publish the peer review history of their article (what does this mean?). If published, this will include your full peer review and any attached files.

Reviewer #1: No

Reviewer #2: No

---

## [Editor Report · Acceptance letter]

13 Sep 2022

PONE-D-22-08667R1 

Motivational factors were more important than perceived risk or optimism for compliance to infection control measures in the early stage of the COVID-19 pandemic 

Dear Dr. Sætrevik:

I'm pleased to inform you that your manuscript has been deemed suitable for publication in PLOS ONE. Congratulations! Your manuscript is now with our production department. 

Kind regards, 

on behalf of

Dr. Ali Safaa Sadiq 

Academic Editor

PLOS ONE